# Copulatory courtship, body temperature and infection in *Tenebrio molitor*

**Franco Cargnelutti**[1,2], **Ulises Castillo-Pérez**[3], **Alicia Reyes-Ramírez**[3], **Maya Rocha-Ortega**[3], **Alex Córdoba-Aguilar**[3]*

1 Departamento de Diversidad Biológica y Ecología, Facultad de Ciencias Exactas, Físicas y Naturales, Universidad Nacional de Córdoba, Córdoba, Argentina, 2 Laboratorio de Biología Reproductiva y Evolución, Consejo Nacional de Investigaciones Científicas y Técnicas (CONICET), Instituto de Diversidad y Ecología Animal (IDEA), Córdoba, Argentina, 3 Departamento de Ecología Evolutiva, Instituto de Ecología, Universidad Nacional Autónoma de México, Coyoacán, México

* acordoba@iecologia.unam.mx

**Data Availability Statement:** Data can be accessed from the figshare repository (https://doi.org/10.6084/m9.figshare.22596052).

## Abstract

Ectothermic animals can raise their body temperature under varying circumstances. Two such situations occur during sexual activity (as metabolic rate rises during copulatory movements) and during infection (to control pathogens more effectively). We have investigated these two situations using *Tenebrio molitor* males. We recorded the copulatory courtship behavior of sick (= infected with *Metharizium robertsii* fungus) vs healthy males and its link with body temperature. We predicted a positive relation between copulatory courtship (measured as antennal and leg contact behavior) and body temperature, especially in sick males. We found that the intensity of contacts correlated with increased body temperature in sick males. Previous studies in this species indicated that partner females laid fewer eggs after mating with sick males above a certain male body temperature threshold. Thus, our present findings suggest that females may detect male infection via intensity of antennal-mediated courtship, body temperature or their combination. If this is the case, females may assess male cues directly related to health status such as body temperature.

## Introduction

Temperature is a key factor in regulating critical biological processes like development, foraging, or reproduction in ectothermic animals whose body temperature is strongly ambient dependent [1–3]. Temperature can influence pre-copulatory courtship [4, 5], copulation duration [6, 7], mate guarding duration [8], sexual communication [9], mating rates [4, 10], and mating strategies [11], among others. Moreover, individuals employ thermoregulatory behaviors to modify their temperature (see [1]). For example, moths may beat or vibrate their wings to warm up their flight muscles before flying, while hornets beat their wings to cool their nests as temperature increases (i.e., transient endothermy) [12, 13].

Body temperature may also be linked to the immune and health status of individuals (e.g., [14]). For example, animals can raise their temperature to an immune optimum to fight off pathogens effectively, a process called behavioral fever [15–17]. Although the literature on the effect of temperature on several behaviors is vast, surprisingly, the copulatory courtship

**Funding:** . This research was financed by a UNAM grant PAPIIT project IN204921. The funders had no role in study design, data collection and analysis, decision to publish, or preparation of the manuscript.

of sick males is an unexplored area. Copulatory courtship is currently considered a rule rather than an exception in sexual interactions [18–20], and is thought to have evolved through cryptic female choice [18, 20, 21]. One unexplored question is whether there is a bidirectional effect between body temperature and copulatory courtship. This missing information can be especially useful to understand patterns of, for example, mate choice. In this case, assessing body temperature can be derived from whether a partner is sick or non-sick [14]. Such assessment would take place via the physical contact that takes place during courtship in many insects [18, 19].

An ideal model organism to study the bidirectional link between copulatory courtship and body temperature in sick ectothermic animals is the beetle *Tenebrio molitor*. Females of this species prefer the pheromones of sick males, yet females lay smaller eggs with lower lipid concentration and reduced hatching success after mating with these males [22]. As copulatory courtship increases (i.e., antennal contacts) in sick males, females invariably end up rearing fewer offspring compared to non-sick males [23]. Sick males use a terminal investment strategy by prioritizing more resources to increase pheromone production at the expense of survival [22]. It has been also suggested that females penalize such terminally investing males, and that male temperature is used as the cue that females use to assess that a male is sick [14]. A recent paper supported this idea: females that copulated with infected males whose temperature exceeded 24°C ended up with an impaired fitness via a reduction in the number of eggs laid [14]. The authors of this work suggest that a higher performance in copulatory courtship of sick males leads to an increase in their body temperature [14, 23]. Thus, our aim in this work is to assess whether females derive information of a male's sick or non-sick health status (from now on, health status) via the intensity of copulatory courtship, body temperature or both. We evaluated the bidirectional nature of copulatory courtship and body temperature in healthy vs sick individuals of *T. molitor*. We predicted: 1) that a male that performs a more intense copulatory courtship, will reach higher body temperatures and 2) infected males will reach a higher body temperature during copulatory courtship compared to healthy males.

## 2. Materials and methods

### 2.1. Insect maintenance

To reduce inbreeding, the colony was formed from five breeding centers of the State of Mexico and Mexico City. The resulting individuals for the fifth generation (approximately) were used for the following experiments. They were kept at 25 ± 2°C with a photoperiod of 12 hours light/12 hours darkness. Each week an apple slice was added as a water source, and wheat bran (Maxilu brand) *ad libitum*. The nutritional content of fiber and protein of this medium (per 30 g) was 12.84 g and 4.68 g, respectively. To reduce cannibalism [24] we kept around 200 larvae in plastic containers (30.5 cm 113 diameter x 10.5 cm height). Pupae were sexed based on the morphology of the eighth abdominal segment [25] and separated to collect virgin individuals for the choice test.

### 2.2. Fungus cultivation and LD50

As an entomopathogen, we used *Metarhizium robertsii* (ARSEF 2134). This fungus was acquired from the Agricultural Research Service of the United States Department of Agriculture. This fungus has been shown to reduce the survival of *T. molitor* [22]. To obtain the $LD_{50}$ (median lethal dose) previously reported [22], spores were transported in a solution of 10% glycerol at -80°C and stored in Sabouraud Dextrose Agar (SDA) for 15 days in incubation at 28°C without exposure to light. Then, conidiophores were collected from the SDA medium and suspended in 0.03% Tween 80 solution (hereafter referred to as Tween). The suspension

was mixed by vortexing for 5 minutes and filtered through a cotton mesh to prevent the inclusion of mycelium. Conidia and their viability were recorded in a Neubauer chamber. With the counting technique on the SDA plate [26], we calculated the conidia's relative viability, which was above 95%. The LD50 (median lethal dose) was obtained from this filtrate, which was previously reported by Reyes-Ramírez et al [22]

## 2.3. Health treatments and application

We varied the health status of the males only three days before tests, once males were sexually mature (see [27]). Males were kept virgin until the day of the test. Note that spores germinate after 24 hrs and invade the insect hemocoel within 3 days [28], causing the males to be sick. Thus, the time given between infection and tests is enough for the males to be sick. However, within this period, the males still do not present symptoms such as lethargy or sporulation of the conidia (All authors unpublished data). The following health status treatments were established: 1) infected males with the entomopathogenic fungus (hereafter, fungus), which were submerged for 5 seconds in a dilution of Tween 80 to 0.03% with conidia at the $LD_{50}$, at an approximate concentration of $3x10^5$ conidia/mL of *M. robertsii;* 2) positive control (hereafter, Tween), whose males were submerged for 5 seconds in 20 mL tween 80 at 0.03% with no conidia, and 3) negative control (hereafter, negative control), whose healthy males were not manipulated. Finally, each individual was placed in 12-well plates and maintained in an incubator at 25°C until the choice test.

## 2.4. Choice assay

Three independent trials of precopulatory female choice were established, in which females evaluated males that varied in health status (Fig 1): 1) negative control vs. Tween; 2) fungus vs. Tween; and 3) fungus vs. negative control. We performed a total of 180 trials using 540 different individuals. All choice assays were performed in a dark room with red light. A Y-olfactometer supplied with an air current was used to carry the volatiles through the entire device. The olfactometer presents gates with pores that allow the passage of volatiles, but no contact between individuals. Each male was randomly placed in one of the two arms while one female was kept at the end of the olfactometer, with previous acclimatization, for 2 minutes. Then the gate was opened, allowing the female to choose one of the two arms [22]. At the end of each trial, the olfactometer was completely cleaned with 70% ethanol to avoid the accumulation of pheromones and chemical residues. Each of the pairs formed remained isolated to allow mating.

## 2.5. Recording of body temperature

Once females chose a male from the Y-olfactometer, we took thermographic photographs of each chosen male using a FLIR® model E6 camera (resolution of 160 x 120 pixels with a spectral range of 7.5–13 μm and a thermal sensitivity of <60 mK at 30°C.) (Fig 1). This was done one minute after copulation started (but while the male was still mounted on the female) and one minute after copulation ended. Photographs were taken approximately 20 cm away from the animals. When the temperature varied between the center of the insect's body to the edge, we used the highest temperature (from the center). The thermal images produced and stored by the thermal camera are made up of color palettes that allow small temperature differences to be visualized. We used the FLIR Tools® software to measure those thermal differences in the core body temperature of individuals (1 mm).

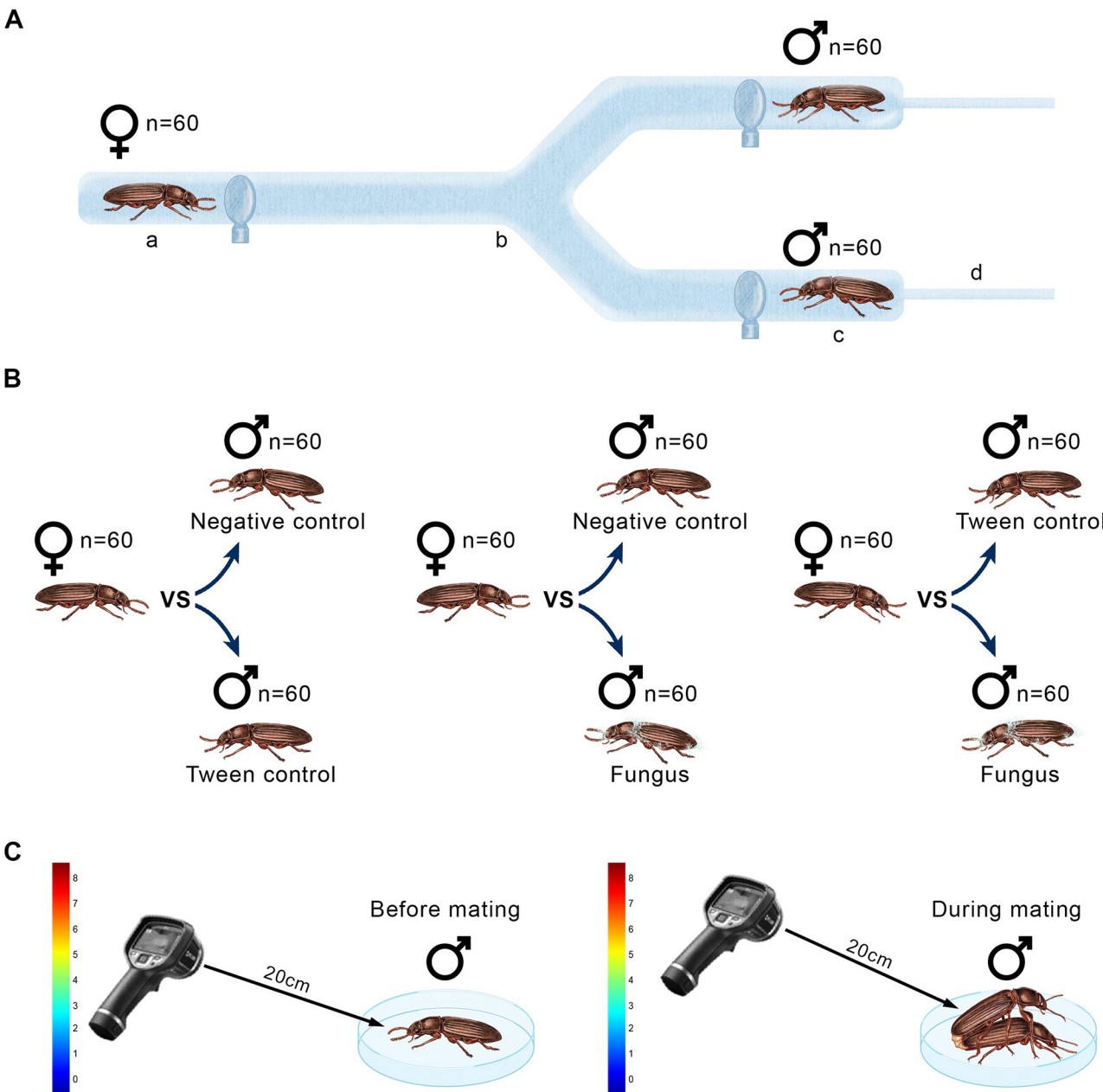

**Fig 1. Schematic representation of the experimental design.** A) Female choice assay using a Y olfactometer (sample sizes are shown): a) Release port where female is placed; b) Section where female makes her choice; c) Arms where males to be chosen are placed; and d) Connections with the air stream; B) The three female choice assays before copulation where the state of health treatments are specified; C) Recording of male temperature according to health status, using a thermal camera before and during copulation.

## 2.6. Recording of copulatory courtship

To record the copulatory behavior of *T. molitor* we followed the protocol in [23]. We placed each pair, female, and the chosen male, in a glass container (12.2 cm diameter × 16.4 cm height) with a fine layer of wheat bran. All copulations were recorded using a 12-megapixel camera (Samsung Galaxy S8+) with an OIS lens. Copulation was considered successful if it lasted more than 30 s, since that is the approximate time that a male takes to transfer a

spermatophore [29]. For the copulation analysis, we considered: a) the number of times the male tapped with his legs on the sides of the female's elytra (hereafter, leg contact behavior); and b) the number of times the male tapped the edges of the elytra and the thorax with his antennae during copulation (hereafter, antennal contact behavior). The person who scored beetle behavior was blind to the male's treatment (infection status). All the experiments were carried out between 12 and 18 h due to *T. molitor's* diurnal habits [30].

## 2.7. Statistical analysis

Analyses were divided into two paths. First, we analyzed the effect of copulatory courtship on male temperature. Second, we analyzed the effect of male temperature on copulatory courtship. To determine if male temperature during copulation (copulatory temperature) or immediately after (postcopulatory temperature) was affected by the male copulatory courtship, we performed an ANCOVA. For each response variable, we tested for the effect of treatment and interaction between treatment and copulatory courtship. If the interaction was not significant, it was removed from the model, so that we were left with the simple effects (see [31]). To avoid overparameterization we performed two models for each response variable. In one model, we tested for the effect of male antennal contact behavior, while in the second model, we tested for the effect of leg contact behavior. We used a robust regression by an M-estimator (RLM) with Huber weighting. We decided to use RLM instead of an ordinary linear model (LM), because the RLM had a better fit. This fit was evaluated by comparing the residual standard error (RSE) between RLM and LM models. Please notice that a robust regression downweights the effects of outliers in our response variable on the analysis. The rlm function of the "MASS" package [32] was used to fit these models and ANOVA tables were obtained using Anova function from the "car" package [33]. If the treatment variable was statistically significant, a posteriori test was carried out using the "emmeans" package [34]. To establish which model would undergo the post hoc test, the RSE between the two models was compared (see results), using the model with a lower RSE value.

To establish whether copulatory courtship (antennal and leg contact behavior) was affected by temperature during and immediately after copulation, we performed generalized linear models (GLMs) with negative binomial error distribution using the *glm.nb* function of the "MASS'" package. For each response variable, we tested for the effect of treatment and the interaction between treatment and temperature. We avoided overparameterizing by performing two models for each response variable. In one model we tested for the effect of male temperature during copulation, while in the other model we tested for the effect of male postcopulatory temperature. As mentioned in the last paragraph, if the treatment variable was statistically significant, a posteriori test was carried out using the "emmeans" package. To establish which model would undergo the post hoc test, the RSE between the two models was compared, using the model with a lower RSE value. All analyses were conducted using Rv.4.0.4 [35].

## 3. Results

Concerning the bidirectional relation between copulatory courtship and male temperature, only the interaction between antennal contact and treatment was statistically significant (Table 1; Fig 2). Temperature increased as fungus males raised the number of antennal contacts. This pattern was strikingly reversed in the negative groups, where their temperature decreased as the number of antennal contacts increased (Fig 2). Finally, no effect of antennal contacts on the temperature of males in the tween group was observed (Fig 2). Concerning the effect of copulatory courtship on postcopulatory temperature, none of the copulatory

**Table 1. Results of RLM models of male temperature during copulation as a function of treatment and copulatory courtship in *Tenebrio molitor* males.**

| Variable | F | DF | p-value |
|---|---|---|---|
| **Male temperature during copulation (First model)** | | | |
| **Antennal** contact behavior | 0.6350 | 1 | 0.4271 |
| Treatment | 0.2421 | 2 | 0.7853 |
| Antennal contact behavior x Treatment | 5.4281 | 2 | **0.0055\*** |
| **Male temperature during copulation (Second model)** | | | |
| Leg contact behavior | 0.7422 | 1 | 0.3907 |
| Treatment | 1.000 | 2 | 0.3706 |

Asterisk indicates significant results p< 0.05.

courtship variables were statistically significant. (Table 2). However, we did find a treatment effect on postcopulatory temperature (Table 2), where fungus males had a higher postcopulatory temperature than males from the negative group. It is important to mention that for this analysis, we used the model with lower RSE (First model < Second model) (Table 3).

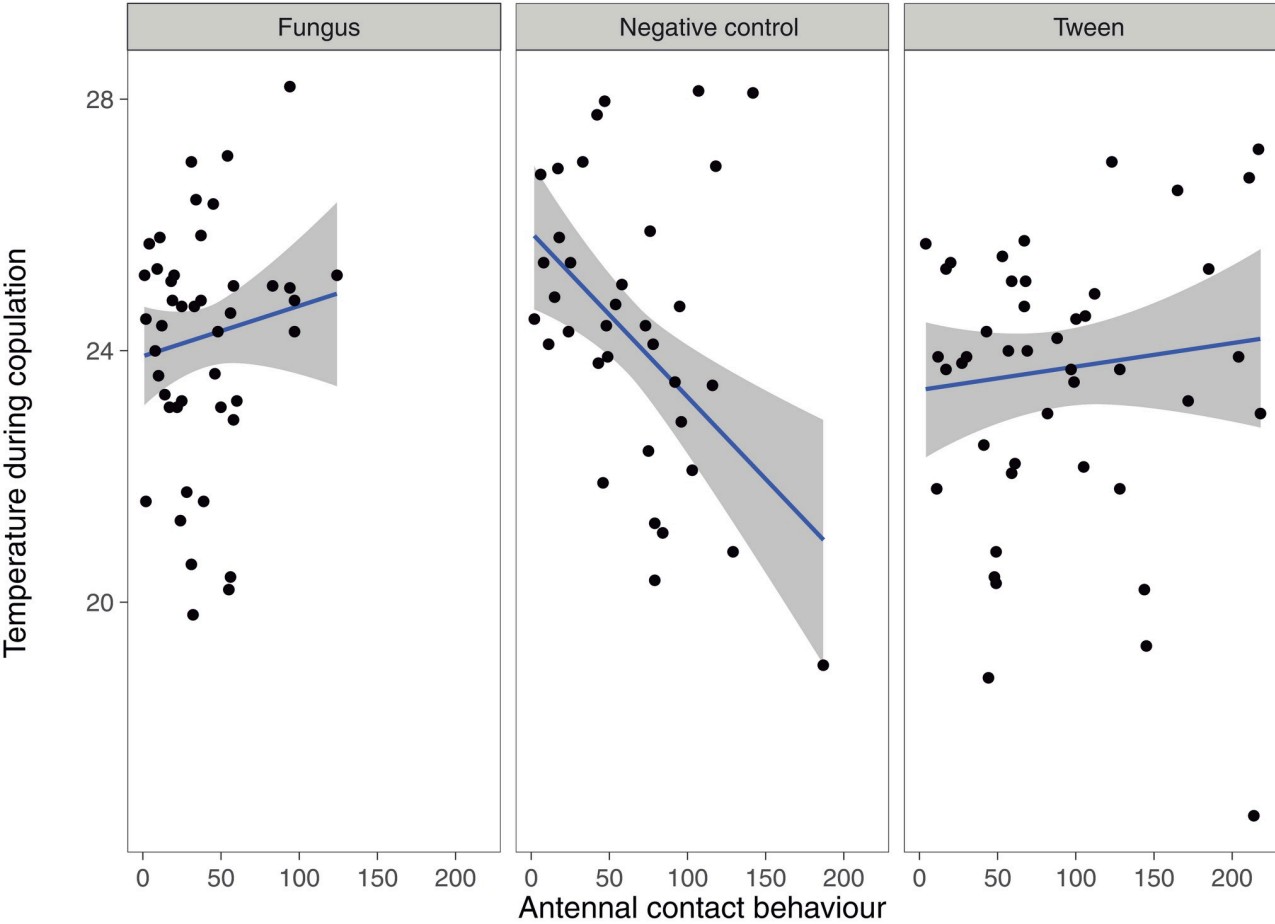

**Fig 2. Male temperature during copulation as a function of treatment and number of antennal contacts in *Tenebrio molitor* males.** Grey bands around the estimated line represent the confidence interval.

**Table 2. Results of RLM models of male temperature after copulation as a function of treatment and copulatory courtship in *Tenebrio molitor* males.**

| Male temperature after copulation (First model) | | | |
|---|---|---|---|
| **Variable** | **F** | **DF** | **p-value** |
| **Antennal** contact behavior | 0.0012 | 1 | 0.9723 |
| Treatment | 3.5283 | 2 | **0.0324*** |
| Male temperature after copulation (Second model) | | | |
| Leg contact behavior | 0.2778 | 1 | 0.5991 |
| Treatment | 3.4791 | 2 | **0.0340*** |

Asterisk indicates significant results p< 0.05.

Regarding the effect of male temperature on male copulatory courtship, we did not find an effect of male temperature during copulation nor of male temperature after copulation on antennal and leg contact behavior (Table 4). However, we did find a treatment effect on the number of antennal contacts, with infected males performing more antennal contacts than tween males (Table 5).

## 4. Discussion

The bidirectional nature between intensity of copulatory courtship and male body temperature is not straightforward in *T. molitor*. Infected males had higher body temperature as the number of antennal contacts increased, while in the negative groups the opposite occurred. One explanation for this is that an increase in body temperature only takes place in this species when an infection occurs. That fever is the response for infections is well known at least for some species [16, 17, 36]. Another hypothesis for this phenomenon is that infected males may have initiated their mating when their temperature was already high. This situation may have occurred when emitting more pheromones than healthy males, a behavior that is common in this species [22, 37, 38]. In this regard, pheromone propagation is known to be temperature dependent (e.g., [39]). Related to this, why did control males show the opposite pattern? It may be that the behaviors that take place during courtship—antennal and leg contact—may serve to cool off rather than to increase temperature. In any case, what does a healthy male gain from reducing his temperature during mating? One explanation may be related to the severity of infections associated with living in a colony. Living in groups conveys the cost of recurrent infections [40–42] and, given this, females may be under constant selection to assess body temperatures of their partners as a way to avoid infection. Concerning this, our unpublished results indicate that females that mate with sick males have a reduced survival (all authors' unpub. data). Thus, it may be that a healthy male is trying to do

**Table 3. Table of contrasts between three levels of the factor treatment of the second model of male temperature after copulation.**

| Contrast | Estimate | SE | z.ratio | p-value |
|---|---|---|---|---|
| Fungus-Negative | 1.020 | 0.419 | 2.434 | **0.0396*** |
| Fungus-Tween | 0.807 | 0.398 | 2.029 | 0.1052 |
| Negative-Tween | -0.213 | 0.423 | -0.503 | 0.8701 |

p-values were adjusted using the Tukey method. Asterisk indicates significant results with an α of 0.05.

**Table 4. Results of GLM models of male copulatory courtship as a function of treatment and male temperature during copulation and after copulation.**

| Variable | χ2 | DF | p-value |
|---|---|---|---|
| **Leg contact behavior (First model)** | | | |
| Temperature during copulation | 1.1555 | 1 | 0.2824 |
| Treatment | 2.0363 | 2 | 0.3613 |
| **Leg contact behavior (Second model)** | | | |
| Temperature after copulation | 0.1512 | 1 | 0.6974 |
| Treatment | 2.6338 | 2 | 0.2680 |
| **Antennal contact behavior (First model)** | | | |
| Temperature during copulation | 0.0484 | 1 | 0.8259 |
| Treatment | 17.7548 | 2 | **0.0001*** |
| **Antennal contact behavior (Second model)** | | | |
| Temperature after copulation | 0.2188 | 1 | 0.6399 |
| Treatment | 17.8663 | 2 | **0.0001*** |

Asterisk indicates significant results p< 0.05.

the opposite to show that he is not sick: reduce their temperature despite relatively intense activity levels. According to this, healthy males may signal their quality to their mates, possibly by investing in physiological/biochemical traits to reduce their temperature. In this fashion, sick males have a clear mismatch in their sexually selected traits (pheromones/courtship) and mate preferences [2]. For females, it would be easier to detect a healthy male when his body temperature is opposite to what a sick male shows. This hypothesis—a reduction in male body temperature to show a healthy state in a colony living animal—awaits further testing.

Sexual interactions in *T. molitor* have indicated that sick or immune-challenged males are more attractive as a likely consequence of increasing pheromone production [22, 37, 38]. Since these males end up reducing their survival, this phenomenon has been interpreted as a terminal investment strategy [22, 38, 43, 44]. However, sick males fathered fewer eggs and/or had a reduced egg hatching success [22, 45]. Our current results can be accommodated based on how females detect sick males. In this fashion, it may be that females that are "lured" to mate with sick males, may correct their mating decisions via sensing different partners' traits such as: a) body temperature; b) copulatory courtship (antennal contact); and/or, c) both. We have already discussed the multimodal nature of sexual signals, as they can indicate several aspects of male condition to females [14]. For the case of *T. molitor*, such information may be related to infection and the inability to keep a certain temperature.

**Table 5. Table of contrasts between three levels of the factor treatment of the second model of male temperature after copulation (model with lower RSE).**

| Contrast | Estimate | SE | z.ratio | p-value |
|---|---|---|---|---|
| Fungus-Negative | -0.394 | 0.181 | -2.169 | 0.0766 |
| Fungus-Tween | -0.731 | 0.181 | -4.260 | **0.0001*** |
| Negative-Tween | -0.337 | 0.184 | -1.833 | 0.1588 |

p-values were adjusted using the Tukey method. Asterisk indicates significant results p< of 0.05.

## Acknowledgments

The authors would like to thank Manuel Sosa for his help editing the figures. We would also like to thank German Gonzalez for his help with the statistical analysis.

## Author Contributions

**Conceptualization:** Franco Cargnelutti, Ulises Castillo-Pérez, Alicia Reyes-Ramírez, Alex Córdoba-Aguilar.

**Data curation:** Franco Cargnelutti, Ulises Castillo-Pérez, Alicia Reyes-Ramírez, Maya Rocha-Ortega, Alex Córdoba-Aguilar.

**Formal analysis:** Franco Cargnelutti, Maya Rocha-Ortega, Alex Córdoba-Aguilar.

**Funding acquisition:** Alex Córdoba-Aguilar.

**Investigation:** Ulises Castillo-Pérez, Alicia Reyes-Ramírez, Alex Córdoba-Aguilar.

**Methodology:** Franco Cargnelutti, Alicia Reyes-Ramírez, Maya Rocha-Ortega, Alex Córdoba-Aguilar.

**Project administration:** Alex Córdoba-Aguilar.

**Resources:** Alex Córdoba-Aguilar.

**Software:** Alex Córdoba-Aguilar.

**Supervision:** Alex Córdoba-Aguilar.

**Validation:** Alex Córdoba-Aguilar.

**Visualization:** Alex Córdoba-Aguilar.

**Writing – original draft:** Franco Cargnelutti, Ulises Castillo-Pérez, Alicia Reyes-Ramírez, Alex Córdoba-Aguilar.

**Writing – review & editing:** Franco Cargnelutti, Alex Córdoba-Aguilar.

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
