## [Decision Letter · Decision Letter 0]

24 Jul 2023

PONE-D-23-11121

Copulatory courtship, body temperature and infection in Tenebrio molitor

PLOS ONE

Dear Dr. Córdoba-Aguilar,

Thank you for submitting your manuscript to PLOS ONE. After careful consideration, we feel that it has merit but does not fully meet PLOS ONE’s publication criteria as it currently stands. Therefore, we invite you to submit a revised version of the manuscript that addresses the points raised during the review process.

We look forward to receiving your revised manuscript.

Kind regards,

Rachid Bouharroud

Academic Editor

PLOS ONE

Journal Requirements:

"FC would like to thank the Consejo Nacional de Investigaciones Científicas y Técnicas (CONICET, Argentina) for financial support. All authors thank the financial support from a UNAM-PAPIIT project IN204921."

"This research was financed by a UNAM grant"

"This research was financed by a UNAM grant"  

Additional Editor Comments:

Dear Authors

As you can see the 2 reviewers decision is major revision. The topic is of interest currently but please address carefully all reviewers comments in order to improve the quality of your manuscript.

Regards

Reviewers' comments:

Reviewer's Responses to Questions

**Comments to the Author**

1. Is the manuscript technically sound, and do the data support the conclusions?

Reviewer #1: Partly

Reviewer #2: Partly

2. Has the statistical analysis been performed appropriately and rigorously? 

Reviewer #1: I Don't Know

Reviewer #2: I Don't Know

3. Have the authors made all data underlying the findings in their manuscript fully available?

Reviewer #1: No

Reviewer #2: Yes

4. Is the manuscript presented in an intelligible fashion and written in standard English?

Reviewer #1: No

Reviewer #2: Yes

5. Review Comments to the Author

Reviewer #1: Reviewer #: Manuscript PONE-D-23-11121 Evaluation Report

Title: " Copulatory courtship, body temperature and infection in Tenebrio molitor"

I would like to thank you for submitting your article titled "Copulatory courtship, body temperature and infection in Tenebrio molitor" to our scientific journal. I have carefully reviewed your research and I am now ready to provide you with my decision regarding its publication.

This study examines how ectothermic animals, particularly male Tenebrio molitor beetles, can increase their body temperature in two situations: during sexual activity and in case of infection. The researchers observed the mating courtship behavior of males infected with the fungus Metharizium robertsii compared to healthy males, and studied the correlation between this behavior and body temperature. They found that the intensity of contacts during mating courtship was associated with an increase in body temperature in sick males. Furthermore, previous studies have shown that females lay fewer eggs after mating with sick males whose body temperature exceeds a certain threshold. The results suggest that females may detect male infection through the intensity of contacts during mating courtship, body temperature, or a combination of both

After a thorough evaluation of your article, Therefore, I recommend that paper could be accepted after major revisions. This decision was made based on my detailed comments and overall assessment of the quality of your work.

Here is a summary of the main comments and concerns I have identified:

Objectives of the article: The objectives of your article are not sufficiently clear, which makes it difficult to understand its scientific contribution

English language quality: I have observed issues with the quality of English in your article, including grammatical errors and difficulties in comprehension that affect the readability of your work.

Abstract: The abstract of your article does not concisely and accurately reflect the main findings of your study, which hinders its clarity and relevance.

The introduction is very well written, the others focus on the difference between cold and warm animals and especially the effect of body temperature on mating as well as the causes of temperature increase in ectothermic animals

Materials and methods: I have identified gaps in the description of the column breeding and experimental manipulations. Some essential information is missing, which impairs the reproducibility of your study.

The breeding conditions used for the tenebrio molitor are not optimal for the breeding, for the minimal temperature for example it is necessary that it is superior 25 °C, if not I am curious to understand if there is a reason behind this choice.

I also ask you to detail the composition of the food used, in particular their protein and fiber content. This significantly affects the reproduction of tenebrio

The monitoring parameters are highly complex and challenging to observe, which negatively affects the rigor of the methods employed in this study.

Results visualization: The graphical representations of your results lack clarity and do not facilitate proper understanding. I encourage improvements in these visualizations, as well as more detailed statistical analyses.

Despite the efforts you have put into your research, I believe it is important to maintain high standards of quality for our scientific journal.

Reviewer #2: This study is somehow interesting, but the experiment can be improved, especially the manipulate of fungus. Here are my comments:

1. Line 104: in the section of “Fungus cultivation and LD50”, I didn’t find a detail of how the authors perform the LD50? How many concentrations did they use for determine the LD50? And the value of LD50 suddenly appeared in the section 2.3.. The authors should provide more detail about this part.

2. Line 117: how do you define the status of “Health”?

3. Line 216: I didn’t agree with the “treatment”. Did the author check the parameters of fungal infection? i.e., the sign or symptom or the reduction of genome copy? Or they found a higher survival rate? There is no data support it.

4. How about use paraffin to block the antenna of female and to see the result of their choic? It just my comment.

6. PLOS authors have the option to publish the peer review history of their article (what does this mean?). If published, this will include your full peer review and any attached files.

Reviewer #1: **Yes: **Dr. Jamaa Zim, PhD

Agricultural engineer

Horticultural Complex in Agadir

Agronomic and veterinary institute Hassan II

BP. 121 86150 Ait-Melloul Agadir, Morocco.

Email: j.zim@iav.ac.ma/zimiavcha@gmail.com

https://www.researchgate.net/profile/Jamaa-Zim

Cell: +212631631946

Reviewer #2: No

---

## [Author Response · Author response to Decision Letter 0]

16 Aug 2023

Dr. Rachid Bouharroud

Academic Editor

PLOS ONE

Dear Dr., Bouharroud,

Thanks very much for your response to our submission. We have produced a new, revised version following the reviewers’ comments. Please use this letter as guidance for checking how we did such revision: use the line numbers or yellow underlining on the revised version.

Thanks very much, and on behalf of my coauthors.

Dr. Alex Córdoba-Aguilar

-------------------

Journal Requirements:

R: Thanks for the comment; we fixed this problem in the new version.

"FC would like to thank the Consejo Nacional de Investigaciones Científicas y Técnicas (CONICET, Argentina) for financial support. All authors thank the financial support from a UNAM-PAPIIT project IN204921."

We note that you have provided additional information within the Acknowledgements Section that is not currently declared in your Funding Statement. Please note that funding information should not appear in the Acknowledgments section or other areas of your manuscript. We will only publish funding information present in the Funding Statement section of the online submission form. Please remove any funding-related text from the manuscript and let us know how you would like to update your Funding Statement. Currently, your Funding Statement reads as follows:

"This research was financed by a UNAM grant"

R: Thanks for the clarification; full funding has been added to the cover letter.

"This research was financed by a UNAM grant" 

R: Thanks for the clarification; we add the statement in the cover letter. 

5. In your Data Availability statement, you have not specified where the minimal data set underlying the results described in your manuscript can be found. PLOS defines a study's minimal data set as the underlying data used to reach the conclusions drawn in the manuscript and any additional data required to replicate the reported study findings in their entirety. All PLOS journals require that the minimal data set be made fully available. For more information about our data policy, please see:

http://journals.plos.org/plosone/s/data-availability.

Upon re-submitting your revised manuscript, please upload your study’s minimal underlying data set as either Supporting Information files or to a stable, public repository and include the relevant URLs, DOIs, or accession numbers within your revised cover letter. For a list of acceptable repositories, please see

http://journals.plos.org/plosone/s/data-availability#loc-recommended-repositories.

Any potentially identifying patient information must be fully anonymized.

Important: If there are ethical or legal restrictions to sharing your data publicly, please explain these restrictions in detail. Please see our guidelines for more information on what we consider unacceptable restrictions to publicly sharing data:

http://journals.plos.org/plosone/s/data-availability#loc-unacceptable-data-access-restrictions. Note that it is not acceptable for the authors to be the sole named individuals responsible for ensuring data access.

R: Thank you for your comment; we have included in the cover letter the DOI of the figshare repository where our data is stored.

Additional Editor Comments:

Dear Authors

As you can see the 2 reviewers decision is major revision. The topic is of interest currently but please address carefully all reviewers comments in order to improve the quality of your manuscript.

Regards

Reviewers' comments:

Reviewer's Responses to Questions

Comments to the Author

1. Is the manuscript technically sound, and do the data support the conclusions?

Reviewer #1: Partly

Reviewer #2: Partly

2. Has the statistical analysis been performed appropriately and rigorously?

Reviewer #1: I Don't Know

Reviewer #2: I Don't Know

3. Have the authors made all data underlying the findings in their manuscript fully available?

Reviewer #1: No

Reviewer #2: Yes

4. Is the manuscript presented in an intelligible fashion and written in standard English?

Reviewer #1: No

Reviewer #2: Yes

 5. Review Comments to the Author

Reviewer #1: Reviewer #: Manuscript PONE-D-23-11121 Evaluation Report

Title: "Copulatory courtship, body temperature and infection in Tenebrio molitor"

I would like to thank you for submitting your article titled "Copulatory courtship, body temperature and infection in Tenebrio molitor" to our scientific journal. I have carefully reviewed your research, and I am now ready to provide you with my decision regarding its publication.

This study examines how ectothermic animals, particularly male Tenebrio molitor beetles, can increase their body temperature in two situations: during sexual activity and in case of infection. The researchers observed the mating courtship behavior of males infected with the fungus Metharizium robertsii compared to healthy males, and studied the correlation between this behavior and body temperature. They found that the intensity of contacts during mating courtship was associated with an increase in body temperature in sick males. Furthermore, previous studies have shown that females lay fewer eggs after mating with sick males whose body temperature exceeds a certain threshold. The results suggest that females may detect male infection through the intensity of contacts during mating courtship, body temperature, or a combination of both

After a thorough evaluation of your article, Therefore, I recommend that paper could be accepted after major revisions. This decision was made based on my detailed comments and overall assessment of the quality of your work.

Here is a summary of the main comments and concerns I have identified:

Q1: Objectives of the article: The objectives of your article are not sufficiently clear, which makes it difficult to understand its scientific contribution.

R1: Thank you for your comment. In this new version, we have clarified the objectives.

Q2: English language quality: I have observed issues with the quality of English in your article, including grammatical errors and difficulties in comprehension that affect the readability of your work.

R2: We appreciate your feedback. In this new version, we have improved the quality of the English.

Q3: Abstract: The abstract of your article does not concisely and accurately reflect the main findings of your study, which hinders its clarity and relevance.

R3: We appreciate your feedback. We have improved the abstract in this new version to comply with the reviewer's suggestion.

Q4: The introduction is very well written; the others focus on the difference between cold and warm animals and especially the effect of body temperature on mating as well as the causes of temperature increase in ectothermic animals.

R4: Thank you for your comment!

Materials and methods: I have identified gaps in the description of the column breeding and experimental manipulations. Some essential information is missing, which impairs the reproducibility of your study.

Q5: The breeding conditions used for the Tenebrio molitor are not optimal for the breeding, for the minimal temperature for example it is necessary that it is superior 25 °C, if not I am curious to understand if there is a reason behind this choice.

I also ask you to detail the composition of the food used, in particular their protein and fiber content. This significantly affects the reproduction of tenebrio

R5: Thank you very much for your observations. We agree that it has been reported that the development time was faster at higher temperatures (for example 29 °C) and a greater number of eggs were obtained. However, at these high temperatures, the weight of pupae and adults decreased. In addition, the fat content does not change significantly when the larvae are reared at 23 to 28 °C. In this study, we used an intermediate temperature (25 °C) in which we know that Tenebrio molitor can develop without problem since it has been used in many other papers before. It was a typo error “20 ºC” now you can find in this version the real value of 25 ºC.

The nutritional value (30 grams) of wheat bran, the medium that the insects had ad libitum, was 12.84 g of fiber and 4.68 g of protein. The rest of the values were for fats, carbohydrates, and sodium. We have specified these values in the main text, "The nutritional content of fiber and protein of this medium (per 30 g) was 12.84 g and 4.68 g, respectively" (lines 105-106).

Q6: The monitoring parameters are highly complex and challenging to observe, which negatively affects the rigor of the methods employed in this study.

R6: Thanks for your comments. If there is any way to enhance clarity, please let us know.

Q7: Results visualization: The graphical representations of your results lack clarity and do not facilitate proper understanding. I encourage improvements in these visualizations, as well as more detailed statistical analyses.

R7: Thank you very much for your comments. However, we respectfully disagree with the reviewer. Considering the type of analysis used in this study, the presented graphs are accurate and appropriate. Additionally, we have provided tables with the obtained results, offering all the available information to the reader. Regarding the statistical details, our analyses have been explained in the utmost detail, outlining how each model was constructed, the type of analysis employed, and the rationale behind each approach. If there is any part of the statistical analysis the reviewer does not understand or feels is unclear, please indicate it in the text, and we will be happy to clarify it.

Despite the efforts you have put into your research, I believe it is important to maintain high standards of quality for our scientific journal.

Reviewer #2: This study is somehow interesting, but the experiment can be improved, especially the manipulate of fungus. Here are my comments:

Q1: Line 104: in the section of “Fungus cultivation and LD50”, I didn’t find a detail of how the authors perform the LD50? How many concentrations did they use for determine the LD50? And the value of LD50 suddenly appeared in the section 2.3.. The authors should provide more detail about this part.

R2: Thank you for your comment. An error in our writing implied that we experimented to calculate the LD50 in this study, which is not the case. This concentration was obtained and reported by Reyes-Ramírez et al. 2019 (a co-author of this work). We have rewritten this section in materials and methods "From this filtrate, the LD50 (median lethal dose) was obtained, which was previously reported by Reyes-Ramírez et al." (lines 120-122).

Q2: Line 117: how do you define the status of “Health”?

R2: “Health” can be broadly understood as whether an animal is fungus-infected (sick) or non-infected (non-sick). This definition appears now in lines 93-95.

Q3: Line 216: I didn’t agree with the “treatment”. Did the author check the parameters of fungal infection? i.e., the sign or symptom or the reduction of genome copy? Or they found a higher survival rate? There is no data support it.

R3: Although we do not measure survival here, this life-history trait has been reported in previous studies (Reyes-Ramírez et al. 2019a, b) where it is shown that the fungus significantly reduces the lifespan of individuals. On the other hand, from 4–5 days after the infection, the sick males begin to present symptoms such as lethargy or sporulation of the fungus, which is why our tests of choice are carried out before the presence of these symptoms. We have added this information to give more clarity and context to our methodology: “This fungus has been shown to reduce the survival of T. molitor” and “However, within this period, the males still do not present symptoms such as lethargy or sporulation of the conidia” (lines 127-130).

4. How about use paraffin to block the antenna of female and to see the result of their choice? It just my comment.

R: A nice idea indeed. We will think about how to implement it in future experiments.

6. PLOS authors have the option to publish the peer review history of their article (what does this mean?). If published, this will include your full peer review and any attached files.

Do you want your identity to be public for this peer review? For information about this choice, including consent withdrawal, please see our Privacy Policy.

Reviewer #1: Yes: Dr. Jamaa Zim, PhD

Agricultural engineer

Horticultural Complex in Agadir

Agronomic and veterinary institute Hassan II

BP. 121 86150 Ait-Melloul Agadir, Morocco.

Email: j.zim@iav.ac.ma/zimiavcha@gmail.com

https://www.researchgate.net/profile/Jamaa-Zim

Cell: +212631631946

Reviewer #2: No

---

## [Editor Report · Decision Letter 1]

29 Aug 2023

Copulatory courtship, body temperature and infection in Tenebrio molitor

PONE-D-23-11121R1

Dear Dr. Córdoba-Aguilar,

We’re pleased to inform you that your manuscript has been judged scientifically suitable for publication and will be formally accepted for publication once it meets all outstanding technical requirements.

Kind regards,

Rachid Bouharroud

Academic Editor

PLOS ONE

Additional Editor Comments (optional):

Dear Dr Cordoba-Aguilar

You manuscript entitled "Copulatory courtship, body temperature and infection in Tenebrio molitor" looks better than the first version and desserve to be published by PlosOne. Please be care to submit the clean version. The edible insect is currently an emerging topic and your findings could be of interest for scientific community and also industrials.

Regards
---

## [Editor Report · Acceptance letter]

1 Sep 2023

PONE-D-23-11121R1 

Copulatory courtship, body temperature and infection in *Tenebrio molitor*

Dear Dr. Córdoba-Aguilar:

I'm pleased to inform you that your manuscript has been deemed suitable for publication in PLOS ONE. Congratulations! Your manuscript is now with our production department. 

Kind regards, 

on behalf of

Dr. Rachid Bouharroud 

Academic Editor

PLOS ONE